

# Charge order and antiferromagnetism in twisted bilayer graphene from the variational cluster approximation

**Bahman Pahlevanzadeh[1,2], Peyman Sahebsara[1] and David Sénéchal[2⋆]**

**1** Department of Physics, Isfahan University of Technology, Isfahan, Iran
**2** Département de physique and Institut quantique, Université de Sherbrooke, Sherbrooke, Québec, Canada J1K 2R1

⋆ david.senechal@usherbrooke.ca

## Abstract

We study the possibility of charge order at quarter filling and antiferromagnetism at half-filling in a tight-binding model of magic angle twisted bilayer graphene. We build on the model proposed by Kang and Vafek [1], relevant to a twist angle of 1.30°, and add on-site and extended density-density interactions. Applying the variational cluster approximation with an exact-diagonalization impurity solver, we find that the system is indeed a correlated (Mott) insulator at fillings $\frac{1}{4}$, $\frac{1}{2}$ and $\frac{3}{4}$. At quarter filling, we check that the most probable charge orders do not arise, for all values of the interaction tested. At half-filling, antiferromagnetism only arises if the local repulsion $U$ is sufficiently large compared to the extended interactions, beyond what is expected from the simplest model of extended interactions.

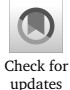

# 1 Introduction

The observation of correlated insulators and superconductivity in twisted bilayer graphene (TBG) [2,3] has inaugurated the new field of twistronics. This discovery was motivated by the prediction that, for a few small "magic" twist angles, the band structure of a twisted graphene bilayer would contain a low-energy manifold of flat bands, well separated from the other bands and forming a strongly correlated electronic subsystem. [4–6]. So far the superconducting order parameter symmetry of TBG is not known, although there are numerous predictions. The precise nature of the insulating state (pure Mott insulator or broken symmetry phase) is not precisely known either. The goal of this paper is to analyze the insulating state of TBG at quarter- and half-filling and to ascertain whether it is a pure Mott state or a broken symmetry state, either a charge-density wave (quarter filling) or an antiferromagnet (half-filling). We will conclude that it is indeed a pure Mott state, within the limits of our model and approximations.

The peculiar properties of magic-angle TBG – insulating behavior, superconductivity or other orders – are of course the effect of interactions. A proper treatment of interactions also depends on the noninteracting description of TBG, and has already been the subject of many studies [7–23]. This paper is an extension of our previous work [24] on the superconducting state of TBG. We will use the same premise: We will start from the four-band, tight-binding effective model proposed by Kang and Vafek [1], based on the microscopic analysis of Moon and Koshino [25]. We will assume that the interaction derives only from on-site Coulomb repulsion at the AA sites, which translates into an extended Hubbard model with the Wannier states living on a honeycomb lattice. However, we will focus here on the normal-state properties. Again, we will analyze the model with a method based on a tiling of the lattice by identical clusters. In order to better capture the dynamical (i.e. non mean-field) effect of the local and extended interactions, we will use a larger, 12-site cluster (or three unit cells), the largest we can solve that has the symmetry of the model. It allows us to treat a fair fraction of the extended interactions within the cluster and thus capture the dynamical correlations. The size of the cluster precludes us from applying cluster dynamical mean field theory (CDMFT) as in Ref. [24], because adding bath degrees of freedom would make the system too large for our exact diagonalization solver. Instead, we will apply another cluster method, the variational cluster approximation (VCA), in which Weiss fields are applied directly to the cluster itself, while keeping the full effect of local interactions. We will extend the VCA by a mean-field treatment of inter-cluster interactions. This mean-field treatment is greatly simplified by the symmetry of the cluster used (all 12 sites are equivalent). Since the model studied is nearly particle-hole symmetric, the conclusions reached at quarter filling also apply at three-quarter filling.

# 2 The low-energy model

Among the various tight-binding Hamiltonian proposed for the low-energy bands of TBG [1, 25–27], we adopt the one described in Ref. [1]. This model features four Wannier orbitals per unit cell (labeled $w_{1,2,3,4}$), with maximal symmetry, on an effective honeycomb lattice, appropriate for a twist angle $\theta = 1.30°$. Each site of the honeycomb lattice is associated with two Wannier orbitals, which it is convenient to imagine located on two different layers, containing respectively the orbitals $w_{1,4}$ and the orbitals $w_{2,3}$. The Wannier orbitals of one layer are schematically illustrated on Fig. 1, borrowed from Ref. [24]. We will only retain the largest hopping integrals among those computed in Ref. [1]; see Table 1 (the notation used is that of Ref. [1]). The most important hopping terms are between Wannier orbitals $w_1$ and $w_4$ and between $w_2$ and $w_3$, i.e., between graphene sublattices, within a given layer. The

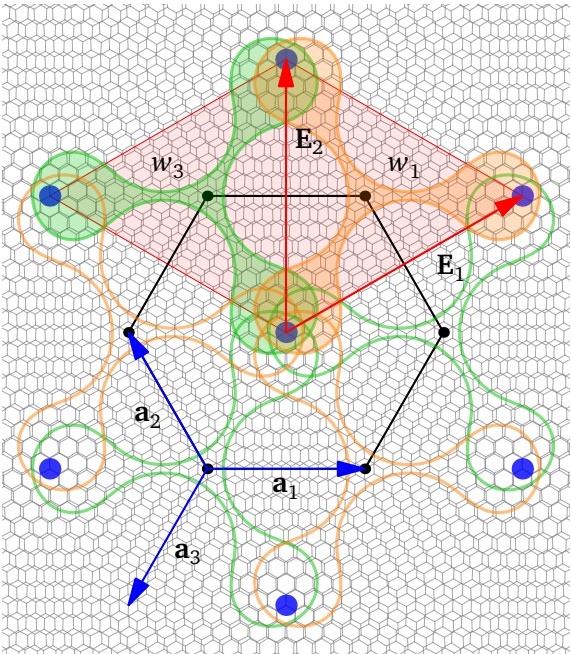

Figure 1: Schematic representation of the Wannier functions $w_1 = w_2^*$ (orange) and $w_3 = w_4^*$ (green) on which our model Hamiltonian is built. The charge is maximal at the AA superposition points (blue circles) forming a triangular lattice. The Wannier functions are centered on the triangular plaquettes that form a graphene-like lattice (black dots), whose unit cell is shaded in red. The basis vectors $\mathbf{E}_{1,2}$ of the moiré lattice are shown (they are also basis vectors of the graphene-like lattice of Wannier functions), as well as the elementary nearest-neighbor vectors $\mathbf{a}_{1,2,3}$. This figure is borrowed from Ref. [24] and is reproduced here for ease of reading.

inter-layer hopping terms are much smaller, the largest of which being $t_{13}[0,0]$. Each of those Wannier orbitals is associated with two degenerate spin projections (the spin-orbit interaction is negligible).

We now proceed to describe a simple model for interactions, derived from an on-site Coulomb repulsion at the AA sites [9, 11, 28]:

$$H_{\text{int}} = u \sum_{\mathbf{R} \in \text{AA}} n_{\mathbf{R}}^2, \tag{1}$$

where the sum is carried over AA sites and $n_{\mathbf{R}}$ is the total charge located at that site, to which contribute 12 Wannier orbitals (6 per layer) and two spin projections per Wannier orbital. Specifically, we could write

$$n_{\mathbf{R}} = \frac{1}{3} \sum_{i=1}^{3} \left( n_{\mathbf{R}+\mathbf{a}_i}^{(1)} + n_{\mathbf{R}-\mathbf{a}_i}^{(1)} + n_{\mathbf{R}+\mathbf{a}_i}^{(2)} + n_{\mathbf{R}-\mathbf{a}_i}^{(2)} \right), \tag{2}$$

where $n_{\mathbf{r}}^{(\ell)} = n_{\mathbf{r}\uparrow}^{(\ell)} + n_{\mathbf{r}\downarrow}^{(\ell)}$ is the total electron number associated with the Wannier orbital centered at the (honeycomb) lattice site $\mathbf{r}$ on layer $\ell$. The vectors $\pm\mathbf{a}_i$, indicated on Fig. 1, go from each AA site to the six neighboring honeycomb lattice sites. The factor of $\frac{1}{3}$ above comes from the fact that each Wannier orbital has three lobes, i.e., is split across three AA sites. Note that we are simply adding the interaction term to the above-defined tight-binding model, in a heuristic way: we have no consistent derivation of the combined model (hopping terms and interactions). A more consistent derivation of the interaction terms can be found in [13]

Table 1: Hopping amplitudes used in the low-energy model, the largest computed in Ref. [1], taken from Ref. [24] and reproduced here for ease of reading. $[a, b]$ stand for the bond vectors in the $(\mathbf{E}_1, \mathbf{E}_2)$ basis of Fig. 1 and $\omega = e^{2\pi i/3}$. The hopping terms $t_{14}$ (full lines) and $t_{23}$ (dashed lines) for a given layer are illustrated on the right; the blue area is the unit cell.

| symbol | value (meV) |
|---|---|
| $t_{13}[0,0] = \omega t_{13}[1,-1] = \omega^* t_{13}[1,0]$ | $-0.011$ |
| • $t_{14}[0,0] = t_{14}[1,0] = t_{14}[1,-1]$ | $0.0177 + 0.291i$ |
| • $t_{14}[2,-1] = t_{14}[0,1] = t_{14}[0,-1]$ | $-0.1141 - 0.3479i$ |
| • $t_{14}[-1,0] = t_{14}[-1,1] = t_{14}[1,-2]$ | |
| $= t_{14}[1,1] = t_{14}[2,-2] = t_{14}[2,0]$ | $0.0464 - 0.0831i$ |

and leads to a departure from the current density-density form of the interaction, which is neglected in this work.

Expressed in terms of the Wannier electron densities $n_{\mathbf{r}}^{\ell}$, the interaction takes the form

$$H_{\text{int}} = \frac{1}{2} \sum_{\mathbf{r},\mathbf{r}',\ell,\ell'} V_{\mathbf{r},\mathbf{r}'}^{\ell,\ell'} n_{\mathbf{r}}^{\ell} n_{\mathbf{r}'}^{\ell'}, \tag{3}$$

where the factor of $\frac{1}{2}$ avoids double counting when performing independent sums over sites and orbitals. The Hubbard on-site, intra-orbital interaction $U$ is equal to $V_{\mathbf{r},\mathbf{r}}^{\ell,\ell}$, since

$$V_{\mathbf{r},\mathbf{r}}^{\ell,\ell} n_{\mathbf{r}\uparrow}^{\ell} n_{\mathbf{r}\downarrow}^{\ell} = \frac{1}{2} V_{\mathbf{r},\mathbf{r}}^{\ell,\ell} (n_{\mathbf{r}\uparrow}^{\ell} + n_{\mathbf{r}\downarrow}^{\ell})(n_{\mathbf{r}\uparrow}^{\ell} + n_{\mathbf{r}\downarrow}^{\ell}) - \frac{1}{2} V_{\mathbf{r},\mathbf{r}}^{\ell,\ell} n_{\mathbf{r}}^{\ell} \qquad (n_{\mathbf{r}\sigma}^2 = n_{\mathbf{r}\sigma}). \tag{4}$$

Including on-site interactions in this form entails a compensation term $U/2$ to the chemical potential.

Careful counting from Eqs (1,2) shows that

$$U = \frac{2}{3}u, \qquad \text{(on-site)}$$

$$V_{\mathbf{rr}}^{(1,2)} \equiv V_0 = \frac{2}{3}u = U, \qquad \text{(same site, different layers)}$$

$$V_{\mathbf{rr}'}^{(\ell,\ell')} \equiv V_1 = \frac{4}{9}u = \frac{2}{3}U, \qquad \text{(1st neighbors)} \tag{5}$$

$$V_{\mathbf{rr}'}^{(\ell,\ell')} \equiv V_2 = \frac{2}{9}u = \frac{1}{3}U, \qquad \text{(2nd neighbors)}$$

$$V_{\mathbf{rr}'}^{(\ell,\ell')} \equiv V_3 = \frac{2}{9}u = \frac{1}{3}U. \qquad \text{(3rd neighbors)}$$

There are no interactions beyond third neighbors coming from a single AA site. We will study this model by assuming the above relations between extended interactions $V_{0,1,2,3}$ and the on-site interaction $U$.

In Ref. [29], constrained RPA estimates of the extended interactions compared to the local interaction are made and applied to lattice models. These estimates depend on the twist angle, and would bring corrections to the constraints (5). From Fig. 10 of Ref. [29], the ratios of 1st and 2nd neighbor interactions to the local interaction at a 1.08° twist angle are 12/28 (instead of 2/3) and 9/28 (instead of 1/3). This is in close agreement for the 2nd neighbor interactions, but 33% off for the first-neighbor interactions. We will nevertheless stick to the constraints (5).

## 2.1 The strong-coupling limit

Given the large number of extended interactions in the model, it is instructive to look at the strong-coupling limit (neglecting all hopping terms) to detect possible charge order instabilities stemming solely from the interactions.

The reader will forgive us if we use a slightly different notation, writing the interaction Hamiltonian as

$$H_{\text{int}} = \frac{1}{2} \sum_{\mathbf{R},\mathbf{R}',a,b} V_{\mathbf{R},\mathbf{R}'}^{a,b} n_{\mathbf{R}}^a n_{\mathbf{R}'}^b, \tag{6}$$

where now $\mathbf{R}$, $\mathbf{R}'$ denote Bravais lattice sites and $a, b$ orbital indices from 1 to 4. In essence, for each $\mathbf{R}$, the site index $\mathbf{r}$ takes two values (the two sublattices $A$ and $B$), as does the layer index $\ell$, leading to four possible value of the orbital index $a$. This shift in notation allows us to express the interaction in Fourier space:

$$H_{\text{int}} = \frac{1}{2} \sum_{\mathbf{q},a,b} \tilde{V}_{\mathbf{q}}^{ab} \tilde{n}_{\mathbf{q}}^{a\dagger} \tilde{n}_{\mathbf{q}}^b, \tag{7}$$

where

$$V_{\mathbf{R}\mathbf{R}'}^{ab} = \frac{1}{L} \sum_{\mathbf{q}} \tilde{V}_{\mathbf{q}}^{ab} e^{i\mathbf{q}\cdot(\mathbf{R}-\mathbf{R}')}, \qquad \tilde{n}_{\mathbf{q}}^a = \frac{1}{\sqrt{L}} \sum_{\mathbf{R}} e^{-i\mathbf{q}\cdot\mathbf{R}} n_{\mathbf{R}}^a. \tag{8}$$

Interactions up to third neighbor are then encoded in the following $\mathbf{q}$-dependent matrix:

$$[\tilde{V}_{\mathbf{q}}^{ab}] = \begin{pmatrix} U + V_2\beta_{\mathbf{q}} & V_1\gamma_{\mathbf{q}} + V_3\gamma_{2\mathbf{q}}^* & V_0 + V_2\beta_{\mathbf{q}} & V_1\gamma_{\mathbf{q}} + V_3\gamma_{2\mathbf{q}}^* \\ V_1\gamma_{\mathbf{q}}^* + V_3\gamma_{2\mathbf{q}} & U + V_2\beta_{\mathbf{q}} & V_1\gamma_{\mathbf{q}} + V_3\gamma_{2\mathbf{q}}^* & V_0 + V_2\beta_{\mathbf{q}} \\ V_0 + V_2\beta_{\mathbf{q}} & V_1\gamma_{\mathbf{q}}^* + V_3\gamma_{2\mathbf{q}} & U + V_2\beta_{\mathbf{q}} & V_1\gamma_{\mathbf{q}} + V_3\gamma_{2\mathbf{q}}^* \\ V_1\gamma_{\mathbf{q}}^* + V_3\gamma_{2\mathbf{q}} & V_0 + V_2\beta_{\mathbf{q}} & V_1\gamma_{\mathbf{q}}^* + V_3\gamma_{2\mathbf{q}} & U + V_2\beta_{\mathbf{q}} \end{pmatrix}, \tag{9}$$

with

$$\beta_{\mathbf{q}} = 2\left(\cos\mathbf{q}\cdot\mathbf{b}_1 + \cos\mathbf{q}\cdot\mathbf{b}_2 + \cos\mathbf{q}\cdot\mathbf{b}_3\right) \quad \text{and} \quad \gamma_{\mathbf{q}} = e^{i\mathbf{q}\cdot\mathbf{a}_1} + e^{i\mathbf{q}\cdot\mathbf{a}_2} + e^{i\mathbf{q}\cdot\mathbf{a}_3}, \tag{10}$$

where the vectors $\mathbf{b}_i$ are the second-neighbor vectors on the honeycomb lattice (hence first neighbors on the Bravais lattice):

$$\mathbf{b}_1 = 2\mathbf{a}_1 + \mathbf{a}_2, \qquad \mathbf{b}_2 = \mathbf{a}_1 + 2\mathbf{a}_2, \qquad \mathbf{b}_3 = \mathbf{a}_2 - \mathbf{a}_1. \tag{11}$$

The order of orbitals adopted in this matrix notation is $(w_1, w_4, w_2, w_3)$: the first two orbitals belong to the "first layer", the last two to the "second layer".

The local density $n_{\mathbf{R}\sigma}^a$ can only take the values 0 or 1, but the Fourier transforms $\tilde{n}_{\mathbf{q}}^a$ are continuous variables in the thermodynamic limit, and they all commute with each other. Hence, for the sake of detecting charge order in the strong-coupling limit, we can treat the variables $\tilde{n}_{\mathbf{q}}^a$ as classical.

The matrix (9) can be diagonalized by a unitary matrix:

$$\tilde{V}_{\mathbf{q}}^{ab} = \sum_{r=1}^{4} U_{\mathbf{q}}^{ar} \lambda_{\mathbf{q}}^{(r)} U_{\mathbf{q}}^{br*} \tag{12}$$

and thus the interaction energy can take the form

$$H_{\text{int}} = \frac{1}{2} \sum_{\mathbf{q}} \sum_{r=1}^{4} \lambda_{\mathbf{q}}^{(r)} |m_{\mathbf{q}}^{(r)}|^2, \qquad \left(m_{\mathbf{q}}^{(r)} = U_{\mathbf{q}}^{ar*} \tilde{n}_{\mathbf{q}}^a\right), \tag{13}$$

with the eigenvalues

$$\lambda_{\mathbf{q}}^{(1)} = U + V_0 + 2V_2\beta_{\mathbf{q}} + 2|V_1\gamma_{\mathbf{q}} + V_3\gamma_{2\mathbf{q}}^*|, \tag{14}$$

$$\lambda_{\mathbf{q}}^{(2)} = U + V_0 + 2V_2\beta_{\mathbf{q}} - 2|V_1\gamma_{\mathbf{q}} + V_3\gamma_{2\mathbf{q}}^*|, \tag{15}$$

$$\lambda_{\mathbf{q}}^{(3)} = \lambda_{\mathbf{q}}^{(4)} = U - V_0. \tag{16}$$

The uniform solution $\tilde{n}_{\mathbf{0}}^a = (1,1,1,1)$ corresponds to $\lambda_{\mathbf{0}}^{(1)}$, which is the largest possible eigenvalue, and is favored by the (neglected) kinetic energy. Since $\tilde{n}_{\mathbf{0}}$ is associated with the total particle number, it cannot vary (unless the chemical potential varies) and thus plays no role in the energetics of charge order. Charge order instabilities in the strong-coupling limit occur for negative eigenvalues, since they can lower the interaction energy. For instance, if $V_{0,2,3}$ were zero, the eigenvalue $\lambda_{\mathbf{q}}^{(2)} = U - 2|V_1\gamma_{\mathbf{q}}|$ would become negative at $\mathbf{q} = 0$ when $V_1 > U/6$, and this would trigger a charge order between the A and B graphene sublattices in the strong-coupling limit. However, when substituting the values given in Eq. (5), one finds that the maximum eigenvalue is $\lambda_{\mathbf{0}}^{(1)} = 12U$ and the minimum eigenvalue is zero, the latter at the Dirac points $\mathbf{q} = \mathbf{K}$ and $\mathbf{q} = \mathbf{K}'$ for $\lambda_{\mathbf{q}}^{(1)}$, and at all wave vectors for $\lambda_{\mathbf{q}}^{(2,3,4)}$. This means that the system has no instabilities in the strong-coupling limit, only indifferent states (zero eigenvalue), especially at wave vectors $\mathbf{K}$ and $\mathbf{K}'$. When probing such instabilities with a cluster method, we should therefore make sure that these two wave vectors belong to the reciprocal cluster. The 12-site (hexagonal) cluster used in this work satisfies this requirement.

## 3 Computational method

### 3.1 The variational cluster approximation

In order to detect spectral gaps in the normal state and to probe the possible existence of antiferromagnetic or charge-ordered states in this model, we use the variational cluster approximation (VCA) [30–32] with an exact diagonalization solver at zero temperature. This method takes into account short-range correlations exactly, while allowing long-range order through the introduction of broken-symmetry fields determined by a variational principle. We will not review this method in detail here, but rather simply summarized the needed procedure; we refer the reader to the literature for details (see, e.g., Refs [33, 34] for example applications).

In VCA, the lattice is tiled into a superlattice of identical clusters. On each cluster one defines a Hamiltonian $H'$, which has the same interaction part as the full Hamiltonian $H$, but a different one-body part $t_{ij}' \neq t_{ij}$ ($i, j$ stand here for site and orbital indices together). The Potthoff functional, inherited from the Luttinger-Ward functional, is a functional of $t_{ij}'$ that is stationary at the physical solution and which takes the following explicit form:

$$\Omega(\mathbf{t}') = \Omega' - \int \frac{d\omega}{2\pi} \sum_{\tilde{\mathbf{k}}} \ln\det\left[\mathbf{1} - \mathbf{W}(\tilde{\mathbf{k}})\mathbf{G}_c(\omega)\right]. \tag{17}$$

This formula deserves a few explanations. We defined the difference $W_{ij} = t_{ij} - t_{ij}'$ of one-body terms between the original and cluster Hamiltonian. We express it as a function of a wave vector $\tilde{\mathbf{k}}$ in the reduced Brillouin zone (associated with the superlattice of clusters) and site/orbital indices within the cluster, which give it its matrix form $\mathbf{W}(\tilde{\mathbf{k}})$. $\mathbf{G}_c(\omega)$ is the electron Green function associated with the Hamiltonian $H'$ on the cluster. $\Omega'$ is the ground state energy (chemical potential included) of $H'$. The frequency integral can be taken along the imaginary axis after proper regularization.

In practice, one looks for stationary points of the functional (17) as a function of a few parameters defining the one-body matrix $\mathbf{t}'$ of the cluster Hamiltonian. Once a solution is found, the cluster self-energy $\Sigma(\omega)$ associated with $H'$ is adopted as the self-energy of the full lattice Hamiltonian: the VCA stems from the exact application of a variational principle on the self-energy, except that the space of available self-energies is limited to the exact self-energies of cluster Hamiltonians parametrized by a finite number of one-body operators. In particular, one can search for spontaneously broken symmetries by including in $\mathbf{t}'$ symmetry-breaking terms, i.e., Weiss fields. By contrast with conventional mean-field theory, the full dynamical effect of correlations is taken into account via the frequency dependence of the cluster Green's function $\mathbf{G}'$ in Eq. (17). In other words, short-range correlations (within the cluster) are treated exactly.

Finally, a few words on the exact diagonalization solver. The ground state of the cluster is computed with the Lanczos method. With a 12-site cluster in the normal phase, the dimension of the Hilbert space is 853,776 in the sector with zero total spin projection at half-filling (12 electrons), and 48,400 at quarter filling. The Green function is computed with the band Lanczos method [35]. This provides a Lehmann representation of the Green function and the latter may be computed at any real or complex frequency.

## 3.2 The dynamical Hartree approximation

The VCA approximation as summarized above only applies to systems with on-site interactions, since the Hamiltonians $H$ and $H'$ must differ by one-body terms only, i.e., they must have the same interaction part. If extended interactions are present, they are partially truncated when the lattice is tiled into clusters and one must apply further approximations. Specifically, we can apply a Hartree (or mean-field) decomposition on the extended interactions that straddle different clusters, while interactions (local or extended) within each cluster are treated exactly. This is called the dynamical Hartree approximation (DHA) and has been used in Ref [36] to study charge order in the extended, one-band Hubbard model and in Refs [33,37] in order to assess the effect of extended interactions on strongly-correlated superconductivity (the qualifier *dynamical* is used to reflect the presence of short-range correlations within the method and its association with methods based on the self-energy, such as VCA or CDMFT). We will explain this approach in this section.

Let us consider a Hamiltonian of the form

$$H = H_0(\mathbf{t}) + \frac{1}{2}\sum_{i,j} V_{ij} n_i n_j, \tag{18}$$

where $i,j$ are compound indices for lattice site and orbital, $n_{i\sigma}$ is the number of electrons of spin $\sigma$ on site/orbital $i$, and $n_i = n_{i\uparrow} + n_{i\downarrow}$ (the index $i$ is a composite of honeycomb site $\mathbf{r}$ and layer $\ell$ indices as used in Sect. 2, or of Bravais lattice site $\mathbf{R}$ and orbital index $a$ used in Sect. 2.1). The factor $\frac{1}{2}$ in the last term comes from the independent sums on $i$ and $j$ rather than a sum over pairs $(i, j)$. In the dynamical Hartree approximation, the extended interactions in the model Hamiltonian (18) are replaced by

$$\frac{1}{2}\sum_{i,j} V^{\mathrm{c}}_{ij} n_i n_j + \frac{1}{2}\sum_{i,j} V^{\mathrm{ic}}_{ij}(\bar{n}_i n_j + n_i \bar{n}_j - \bar{n}_i \bar{n}_j), \tag{19}$$

where $V^{\mathrm{c}}_{ij}$ denotes the extended interaction between orbitals belonging to the same cluster, whereas $V^{\mathrm{ic}}_{ij}$ those interactions between orbitals of different clusters. Here $\bar{n}_i$ is a mean-field, presumably the average of $n_i$, but not necessarily, as we will see below. Both the first term ($\hat{V}^{\mathrm{c}}$) and the second term ($\hat{V}^{\mathrm{ic}}$), which is a one-body operator, are part of the lattice Hamiltonian $H$ and of the VCA reference Hamiltonian $H'$.

Let us express the index $i$ as a cluster index $c$ and a site-within-cluster index $\alpha$. Then Eq. (19) can be expressed as

$$\frac{1}{2}\sum_{c,\alpha,\beta}\tilde{V}^{c}_{\alpha\beta}n_{c,\alpha}n_{c,\beta} + \frac{1}{2}\sum_{c,\alpha,\beta}\tilde{V}^{ic}_{\alpha\beta}(\bar{n}_{\alpha}n_{c,\beta} + n_{c,\alpha}\bar{n}_{\beta} - \bar{n}_{\alpha}\bar{n}_{\beta}), \tag{20}$$

where we have assumed that the mean fields $\bar{n}_i$ are the same on all clusters, i.e., they have minimally the periodicity of the superlattice, hence $\bar{n}_i = \bar{n}_\alpha$. We have consequently replaced the large, $N \times N$ and block-diagonal matrix $V^c_{ij}$ by a small, $N_c \times N_c$ matrix $\tilde{V}^c_{\alpha\beta}$, and we have likewise "folded" the large $N \times N$ matrix $V^{ic}_{ij}$ into the $N_c \times N_c$ matrix $\tilde{V}^{ic}_{\alpha\beta}$.

In order to make this last point clearer, let us consider the simple example of a one-dimensional lattice with nearest-neighbor interaction $v$, tiled with 3-site clusters. The interaction Hamiltonian

$$H_{\text{int}} = v\sum_{i=0}^{N} n_i n_{i+1} \tag{21}$$

would lead to the following $3 \times 3$ interaction matrices:

$$\tilde{V}^c = v\begin{pmatrix} 0 & 1 & 0 \\ 1 & 0 & 1 \\ 0 & 1 & 0 \end{pmatrix}, \qquad \tilde{V}^{ic} = v\begin{pmatrix} 0 & 0 & 1 \\ 0 & 0 & 0 \\ 1 & 0 & 0 \end{pmatrix}. \tag{22}$$

In practice, the symmetric matrix $\tilde{V}^{ic}_{\alpha\beta}$ is diagonalized and the mean-field inter-cluster interaction is expressed in terms of eigen-operators $m_\mu$:

$$\hat{V}^{ic} = \sum_{\mu} D_\mu\left[\bar{m}_\mu m_\mu - \frac{1}{2}\bar{m}_\mu^2\right]. \tag{23}$$

For instance, in the above simple one-dimensional problem, these eigen-operators $m_\mu$ and their corresponding eigenvalues $D_\mu$ are

$$\begin{aligned} D_1 &= -v, & m_1 &= (n_1 - n_3)/\sqrt{2}, \\ D_2 &= \phantom{-}0, & m_2 &= n_2, \\ D_3 &= \phantom{-}v, & m_3 &= (n_1 + n_3)/\sqrt{2}. \end{aligned} \tag{24}$$

The mean fields $\bar{n}_i$ are determined either by applying (i) self-consistency or (ii) a variational method. In the case of ordinary mean-field theory, in which the mean-field Hamiltonian is entirely free of interactions, these two approaches are identical. In the present case, where the mean-field Hamiltonian also contains interactions treated exactly within a cluster, self-consistency does not necessarily yield the same solution as energy minimization. In the first case, the assignation $\bar{n}_i \leftarrow \langle n_i \rangle$ would be used to iteratively improve on the value of $\bar{n}_i$ until convergence. In the second case, one could treat $\bar{n}_i$ like any other Weiss field in the VCA approach, except that $\bar{n}_i$ is not defined only on the cluster, but on the whole lattice. We will follow the latter approach below.

## 4 The normal state at quarter filling

In this work we use a 12-site cluster containing 3 unit cells of the low-energy model. It is made of two superimposed hexagonal clusters, as illustrated on Fig. 2. On that figure the various extended interactions $V_0$ to $V_3$ are indicated. The three wavevectors of the reciprocal cluster

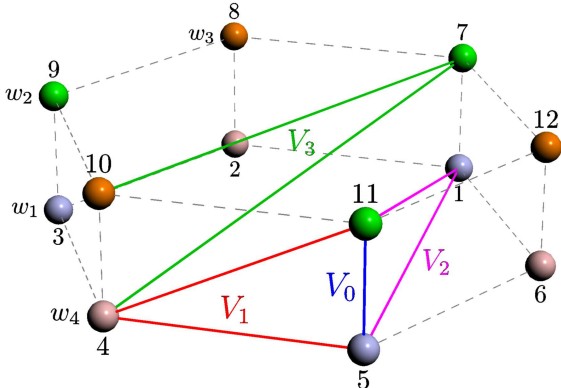

Figure 2: 12-site cluster used in this work. The extended interactions $V_0$ to $V_3$ are shown. Different Wannier orbitals are shown as spheres of different colors. Orbitals $w_1$ and $w_4$ are located, say, on the bottom layer, whereas orbitals $w_2$ and $w_3$ are located on the top layer.

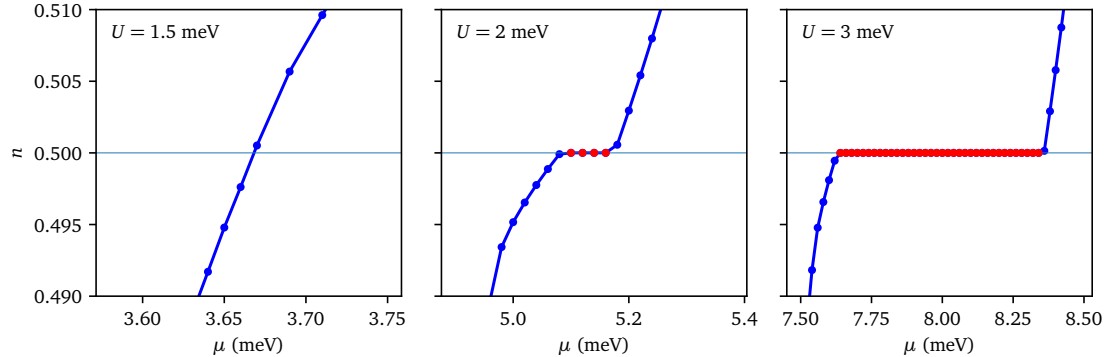

Figure 3: Electronic density vs chemical potential $\mu$ for different interaction strengths at quarter filling. The presence of a plateau (in red) is the signature of an insulating state, and the width of the plateau is the magnitude of the gap. The insulator-to-metal transition occurs between $U = 1.5$ meV and $U = 2$ meV.

are $\mathbf{\Gamma} = \mathbf{0}$, $\mathbf{K}$ and $\mathbf{K}'$. The $12 \times 12$ matrix of inter-cluster interactions is given in Table 2 and the eigen-operators $m_\mu$ used in the dynamical Hartree approximation are illustrated in the lower part of the same table.

We begin by investigating the normal state at quarter filling, for several values of the interaction $U$, all the extended interactions following from $U$ according to Eq. (5). We will start by applying VCA to detect the insulating state, assuming that no charge order is present. To do this, we treat the cluster chemical potential, $\mu_c$, as the sole variational parameter in the VCA procedure. We do not take into account inter-cluster interactions, i.e., the Hartree approximation described in Sect. 3.2. Indeed, all the sites of the 12-site cluster are equivalent in the absence of charge order, meaning that the relevant (normalized) eigenvector of the inter-cluster interaction matrix $V^{\mathrm{ic}}$ is

$$m_0 = \frac{1}{2\sqrt{3}} \sum_{i=1}^{12} n_i \,. \tag{25}$$

Therefore, adding the corresponding mean-field $\bar{m}_0 m_0$ to the lattice Hamiltonian would simply shift the chemical potential by $-\bar{m}_0$, and leave the variational space used in VCA unchanged. This would therefore not help us in determining whether there is a gap or not.

Table 2: Inter-cluster coupling matrix for the 12-site cluster used in this work. The numbering of sites is illustrated on Fig. 2. Bottom: eigenvalues $D_\mu$ and corresponding eigenvectors (or eigen-operators) $m_\mu$ of this matrix. The eigen-operators are shown graphically as a function of site on the 12-site cluster: blue means 1 and red $-1$. The eigenvalues are also shown as a function of the on-site repulsion $U$ when the constraints (5) are applied.

$$\tilde{V}^{\text{ic}} = \begin{pmatrix} 0 & V_3 & 2V_2 & V_1 & 2V_2 & V_3 & 0 & V_3 & 2V_2 & V_1 & 2V_2 & V_3 \\ V_3 & 0 & V_3 & 2V_2 & V_1 & 2V_2 & V_3 & 0 & V_3 & 2V_2 & V_1 & 2V_2 \\ 2V_2 & V_3 & 0 & V_3 & 2V_2 & V_1 & 2V_2 & V_3 & 0 & V_3 & 2V_2 & V_1 \\ V_1 & 2V_2 & V_3 & 0 & V_3 & 2V_2 & V_1 & 2V_2 & V_3 & 0 & V_3 & 2V_2 \\ 2V_2 & V_1 & 2V_2 & V_3 & 0 & V_3 & 2V_2 & V_1 & 2V_2 & V_3 & 0 & V_3 \\ V_3 & 2V_2 & V_1 & 2V_2 & V_3 & 0 & V_3 & 2V_2 & V_1 & 2V_2 & V_3 & 0 \\ 0 & V_3 & 2V_2 & V_1 & 2V_2 & V_3 & 0 & V_3 & 2V_2 & V_1 & 2V_2 & V_3 \\ V_3 & 0 & V_3 & 2V_2 & V_1 & 2V_2 & V_3 & 0 & V_3 & 2V_2 & V_1 & 2V_2 \\ 2V_2 & V_3 & 0 & V_3 & 2V_2 & V_1 & 2V_2 & V_3 & 0 & V_3 & 2V_2 & V_1 \\ V_1 & 2V_2 & V_3 & 0 & V_3 & 2V_2 & V_1 & 2V_2 & V_3 & 0 & V_3 & 2V_2 \\ 2V_2 & V_1 & 2V_2 & V_3 & 0 & V_3 & 2V_2 & V_1 & 2V_2 & V_3 & 0 & V_3 \\ 0 & 2V_2 & V_1 & 2V_2 & V_3 & 0 & V_3 & 2V_2 & V_1 & 2V_2 & V_3 & 0 \end{pmatrix}$$

$m_0$  $m_1$  $m_2$  $m_3$

$D_0 = 2(V_1 + 4V_2 + 2V_3) = \frac{16}{3}U$    $D_1 = 2(V_1 - 2V_2 - V_3) = -\frac{2}{3}U$    $D_2 = 2(V_1 - 2V_2 - V_3) = -\frac{2}{3}U$    $D_3 = -2(V_1 + 2V_2 - V_3) = -2U$

$m_4$  $m_5$  $m_6$  $m_7$

$D_4 = -2(V_1 + 2V_2 - V_3) = -2U$    $D_5 = -2(V_1 - 4V_2 + 2V_3) = 0$    $D_6 = 0$    $D_7 = 0$

$m_8$  $m_9$  $m_{10}$  $m_{11}$

$D_8 = 0$    $D_9 = 0$    $D_{10} = 0$    $D_{11} = 0$

The signature of the Mott gap will be a plateau in the relation between $\mu$ and the density $n$. This is shown in Fig. 3 for a few values of the interaction $U$. Using the cluster chemical potential $\mu_c$ as a variational parameter makes the plateaux very sharp, whereas not using VCA, i.e., simple cluster perturbation theory (CPT) would make the plateaux softer, thereby making the transition to the metallic state more difficult to detect. In the case shown, the metal-insulator transition clear occurs between $U = 1.5$ meV and $U = 2$ meV. This Mott transition is essentially caused by extended interactions: if we imagine a quarter-filled, ground state configuration in the strong coupling limit, with an electron located at every other site (there are many such configurations), adding an extra electron will entail an energy cost proportional to $U$ because of these extended interactions, even though the effect is not as intense as at half-filling.

The question then arises as to the nature of the insulating state at quarter filling: is there

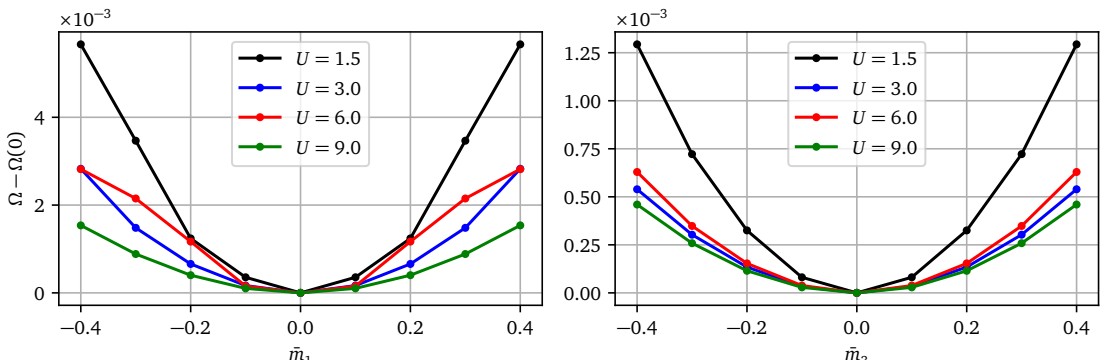

Figure 4: Left panel : The Potthoff functional $\Omega$ as a function of the charge-density-wave Weiss field $\bar{m}_1$ at quarter-filling. Right panel: the same, for the charge-density-wave Weiss field $\bar{m}_3$. See Table 2 for an illustration of the density-waves $m_1$ and $m_3$. The symmetric state (no charge density wave) $\bar{m}_{1,3} = 0$ is stable. All quantities ($U$, $\Omega$, Weiss fields) are in meV.

a charge density wave or not? As shown in Sect. 2.1, the charge fluctuations are expected to be large, because a full array of charge configurations do not affect the energy in the strong-coupling limit when the extended interactions follow Eq. (5). We do expect, on intuitive grounds, that the kinetic energy terms would be unfavorable to charge order. Nevertheless, in order to probe the possible existence of charge order, we will apply Hartree inter-cluster mean-field theory, as described in Sect. 3.2. In order to put all the chances on our side, we will probe one of the eigen-operators with the lowest (negative) eigenvalues in Table 2, namely one of those with $D = -2U$:

$$m_3 = \frac{1}{2\sqrt{2}}\left(n_1 - n_3 - n_4 + n_6 + n_7 - n_9 - n_{10} + n_{12}\right). \tag{26}$$

We must then optimize the Potthoff functional as a function of the mean field $\bar{m}_3$, in addition to using $\mu_c$ as a variational parameter. On the right panel of Fig. 4 we show the Potthoff functional $\Omega$ as a function of $\bar{m}_3$ for a value $\mu_c$ that actually optimize $\Omega$ at a value of $\mu$ associated with quarter filling, for a few values of the interaction $U$. This is to illustrate the absence of nontrivial solution for $\bar{m}_3$, i.e., the value of the mean-field parameter $\bar{m}_3$ that minimizes the energy is indeed zero. This shows that, within this inter-cluster mean-field approximation and for these values of $U$, there is no charge order this type ($m_3$ or, equivalently, $m_4$) at quarter-filling.

We perform the same computation for the $m_1$ eigen-operator:

$$m_1 = \frac{1}{2\sqrt{2}}\left(n_1 - n_3 + n_4 - n_6 + n_7 - n_9 + n_{10} - n_{12}\right) \tag{27}$$

and find similar results, as shown on the left panel of Fig. 4. Therefore, for the values of $U$ probed, the quarter-filled state appears to be a pure, uniform Mott insulator, driven by extended interactions.

We also checked that the self-consistent approach (see paragraph after Eq. (24)) also yields a null result, i.e., no charge order.

Some studies have hinted at the ferromagnetic nature of the quarter-filled state (see, e.g., Ref. [15]). Indeed, ferromagnetism has been likely detected in twisted bilayer graphene with a twist angle of about 1.20° at three-quarter filling [38]. Carrying out VCA computations for

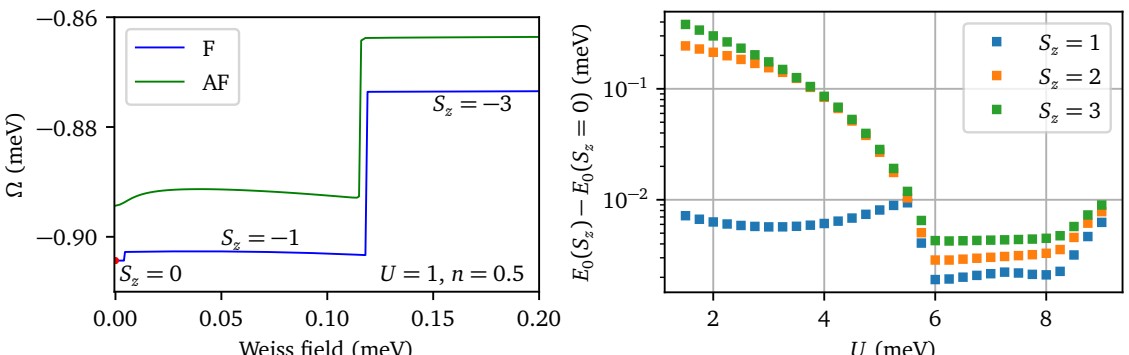

Figure 5: Left panel, blue curve: Potthoff functional vs the ferromagnetic Weiss field $F'$ for $U = 1$ meV, at quarter-filling. The curve is made of distinct continuous pieces, each associated with a value of the total spin projection $S_z$ on the cluster. The minima do not have zero derivative. The only solution *stricto sensu* is at $F' = 0$ (red dot). The green curve is the corresponding result for a Weiss field that is still ferromagnetic within layers, but antiferromagnetic between layers (it is shifted up by 0.01 meV for clarity); it has a smooth minimum at zero. Right panel: the spectral gap between the lowest-energy states for different values of total spin projection $S_z$ and the ground state at $S_z = 0$ for the quarter-filled cluster, as a function of $U$. Note the logarithmic scale. The spectral gap becomes very small after $U \sim 6$ meV, hinting at an instability towards ferromagnetism. In each case the chemical potential $\mu$ and its cluster value $\mu'$ were set so as to sit precisely at quarter filling.

ferromagnetism requires extra care because the corresponding Weiss field

$$\hat{F}' = F' \sum_{i=1}^{12} (n_{i\uparrow} - n_{i\downarrow}) \tag{28}$$

(the sum is over cluster sites) is a conserved quantity (the total spin projection). Thus, the cluster ground state changes discontinuously as $F'$ increases from zero, producing a Potthoff functional profile $\Omega(F')$ that is piecewise continuous. The blue curve on the left panel of Fig. 5 shows a typical profile of the Potthoff functional $\Omega$ at quarter filling (at $U = 1$ meV in that case). A variation on this is to assume ferromagnetic order within layers, with the two layers ordered antiferromagnetically one against the other; in that case there is not net spin projection. The green curve in the same figure shows the Potthoff functional in that case, with a minimum at zero and a similar jump later on.

As the ferromagnetic Weiss field $F'$ is increased, the Zeeman effect shifts the ground state across different total spin sectors (at $U = 1$, it goes directly from $S_z = -1$ to $S_z = -3$, bypassing $S_z = -2$; for larger values of $U$ this is not so, but at the same time the $S_z = 0$ becomes narrower and narrower, and the $S_z = -3$ sector more and more dominant). The discontinuities in $\Omega$ lead to minima without zero derivative, which do not count as VCA solutions. The only solutions are maxima, and the lowest valued maximum is at $F = 0$. Therefore, in the strict sense of VCA, for this cluster and set of variational parameters ($\mu'$ and $F'$), there is no ferromagnetic solution. However, as the right panel of Fig. 5 shows, the energy difference between different spin sectors at quarter filling is small, and becomes smaller as $U$ increases, especially after $U = 6$ meV. That figure hints at some ferromagnetic phase transition around $U \sim 6$ meV, which might be revealed with a larger cluster.

Although the effect of interactions in models of magic-angle TBG has been investigated before, the variety of effective models and methods used makes a comparison difficult. For instance, the effect of interactions in a purelyl microscopic model [14, 17] reveals critical local

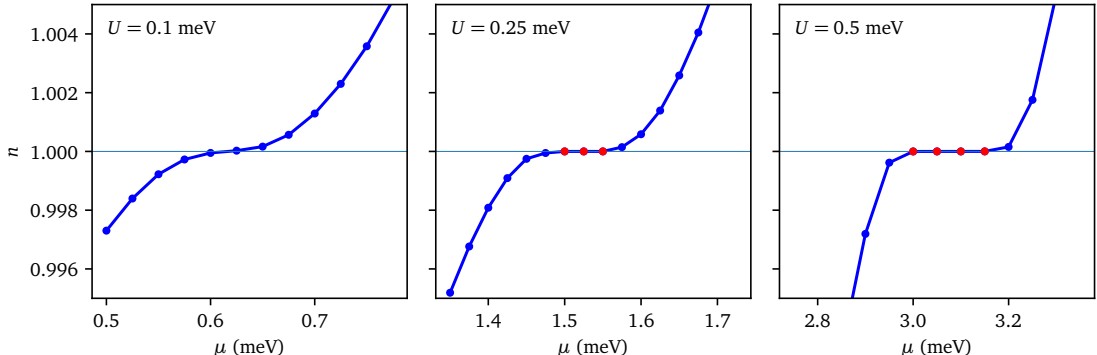

Figure 6: Electronic density vs chemical potential $\mu$ for different interaction strengths at half filling, similar to Fig. 3. The presence of a plateau (in red) is the signature of an insulating state, and the width of the plateau is the magnitude of the gap. The insulator-to-metal transition occurs between $U = 0.1$ meV and $U = 0.25$ meV.

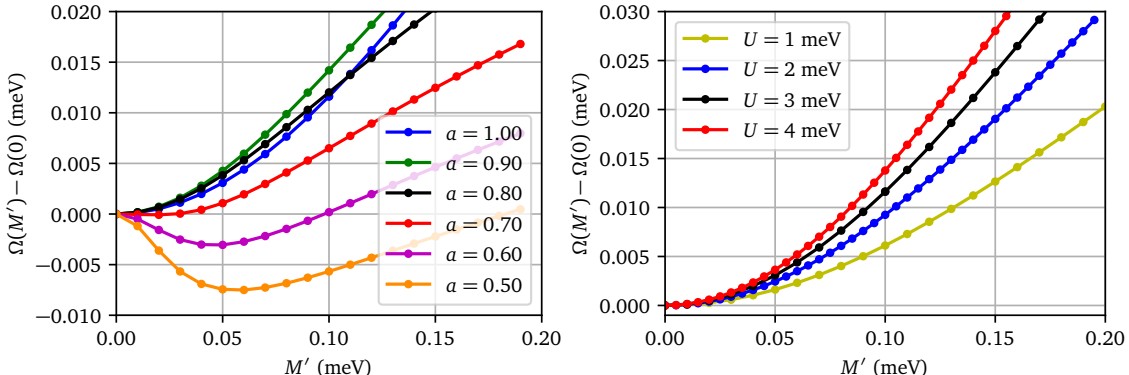

Figure 7: Left panel: Potthoff functional vs the antiferromagnetic Weiss field $M'$ for several values of $a = 3V_1/2U$ and $U = 3$ meV, at half-filling. The case $a = 1$ corresponds to the constraints (5), and smaller values of $a$ just weaken the extended interactions compared to the on-site interaction. The value of $\Omega$ at $M = 0$ is subtracted for clarity. Antiferromagnetism appears only below $a = 0.7$, i.e., not for the extended interactions constrained by Eq. (5). Right panel: same thing, but for $a = 1$ and different values of $U$. Antiferromagnetism does not occur in the range of $U$ studied.

$U$'s in the electron-volt range, but translating this into interactions within an effective moiré model is far from obvious. Local magnetic textures within microscopic models were also investigated [21], but again this does not translate simply in terms of an effective, low-energy model.

## 5  The normal state at half filling and antiferromagnetism

The insulating state at half-filling is revealed the same way as at quarter-filling, by applying the VCA with $\mu_c$ as a variational parameter. The results are shown in Fig. 6, where it appears that the Mott transition occurs between $U = 0.1$ meV and $U = 0.25$ meV, i.e., at a much lower value of the interaction than at quarter filling.

We will not probe charge order at half-filling, as an antiferromagnetic state is more ex-

pected to occur. The Weiss field used to probe antiferromagnetism is

$$\hat{M}' = M' \sum_{i=1}^{12} (-1)^i (n_{i\uparrow} - n_{i\downarrow}),$$

(29)

where $i$ labels sites on the cluster, as defined in Fig. 2. On the infinite lattice, this corresponds to Néel antiferromagnetism within a layer, but ferromagnetism between the two layers. We checked that changing the sign of the staggered magnetization on the second layer, i.e. defining a fully antiferromagnetic operator in all directions, does not affect the results, owing to the small value of the inter-layer hopping in the model. The left panel of Fig. 7 shows the Potthoff functional as a function of $M'$ for different values of the extended interactions compared to the on-site repulsion $U = 3$ meV. These different values are characterized by the ratio $a = 3V_1/2U$, which is unity when the extended interactions obey the constraints (5). Otherwise, the extended interactions $V_{0-3}$ have the same ratios between them as in Eq. (5). Lower values of $a$ correspond to weaker extended interactions (compared to $U$). From that figure we see that, even at a relatively strong $U$ (the Mott transition occurs at a much lower value of $U$), antiferromagnetism is not present at half-filling for the nominal values of the extended interactions defined in Eq. (5). Upon lowering these interactions, antiferromagnetism appears. The right panel of Fig. 7 shows the same type of data for $a = 1$ and different values of $U$, showing that the absence of antiferromagnetism extend to lower and higher values of $U$, the tendency being that it is less and less favored as $U$ increases. Hence the half-filled state should be a true Mott insulator, not an antiferromagnetic insulator.

This is relatively easy to understand in the strong-coupling limit, when Eq. (5) holds. The low-energy manifold at half-filling in the absence of hopping terms is degenerate not only because of spin, but also because of charge motion: if there is exactly one electron on each site, hopping an electron to the neighboring site does not change the interaction energy, and thus the usual strong-coupling perturbation theory argument leading to an effective Heisenberg model at half-filling and large $U$ does not hold anymore.

## 6 Conclusion

We have probed the insulating states at quarter- and half-filling in a tight-binding model for magic angle twisted bilayer graphene, augmented with local and extended density-density interactions. For a wide range of interactions obeying the constraints (5), we have detected the Mott gap using the variational cluster approximation (VCA) with a 12-site cluster and located the Mott transition between $U = 1.5$ meV and $U = 2$ meV at quarter filling, and between $U = 0.1$ meV and 0.25 meV at half-filling. In addition, we have investigated the possibility of charge order at quarter-filling using the VCA and an inter-cluster Hartree approximation for the extended interactions, and concluded that it does not arise. Lastly, we have probed antiferromagnetism at half-filling and concluded likewise that it does not arise when the extended interactions obey the relations (5). It looks therefore plausible that the correlated insulating states observed at these filling ratios are genuine Mott insulators and not gapped ordered states.

**Funding information** DS acknowledges support by the Natural Sciences and Engineering Research Council of Canada (NSERC) under grant RGPIN-2020-05060. Computational resources were provided by Compute Canada and Calcul Québec.

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
