# Peer review of "Charge order and antiferromagnetism in twisted bilayer graphene from the variational cluster approximation"

_SciPost Physics, doi:SciPost Phys. 13, 040 (2022)_

## Round 1 · Referee Report · Anonymous (Referee 1) · 2022-1-14

Strengths

The authors use a cluster approach treating short-range interactions within the cluster exactly.

Weaknesses

The description of the model is insufficient.

The references are insufficient.

Not all possible charge and spin orders are analyzed. A combination of charge and spin order is not analyzed.

Because of using an effective model, order inside one unit cell of the Moire lattice is not considered. This could lead to the possibility of a local antiferromagnetic order inside the unit cell and a ferromagnetic order between different unit cells. Such possibilities are not analyzed.

Report

The authors analyze correlated insulating states in an effective model for twisted bilayer Graphene using variational cluster approximation. They include nonlocal density-density interactions arising from the extended nature of the Wannier orbitals. To include these nonlocal interactions, they use the Hartree approximation within the VCA. They find that the insulating state at quarter filling for large enough interaction strength is a Mott state without long-range charge order. Furthermore, they find that an antiferromagnetic state only arises at half-filling if the nonlocal interactions are sufficiently weak compared to the local interaction. Otherwise, the insulating state at half-filling is a Mott state without long-range order.

Using cluster methods to analyze correlations for an effective model of twisted bilayer Graphene, the authors draw some interesting conclusions for this material.
While I would generally recommend publication, there are several issues that must be changed before publication.

Requested changes

(1) Although there are many publications about correlation effects in twisted bilayer Graphene in experiment and theory, the authors cite only 18 publications. This must be expanded.
A short and not extensive list would be
PhysRevX.8.031089
PhysRevB.98.081102
J. Phys. Commun. 3 035024
PhysRevLett.124.097601
PhysRevB.102.035136
PhysRevB.102.045107
PhysRevB.102.085109
SciPostPhys.11.4.083
PhysRevB.100.155145

As several of these papers also discuss Mott states and antiferromagnetic states, the current results should be compared to these previous results.

(2) During the explanation of the model, I am missing the spin degrees of freedom, which suddenly appear in equation 4. This should be expanded. Furthermore, the condition on the spin degrees should be stated in the strong-coupling section and the calculations for quarter-filling.

(3) Maybe I misunderstood something in the calculation of the strong-coupling limit. When the largest eigenvalue corresponds to the uniform state, then the lowest eigenvalues correspond to nonuniform (charge-ordered) states. Why is the ground state in the strong-coupling limit (when neglecting the kinetic energy) not long-range ordered?

(4) In the section about VCA, the details about how the finite cluster has been solved are missing. This should be expanded.

(5) The authors analyze at quarter-filling only charge order and at half-filling only antiferromagnetism. Why is there no analysis of combinations of charge and spin order? Furthermore, other long-range spin orders other than antiferromagnetism should be discussed.
If possible, the authors should include combined spin and charge order calculations in this manuscript.

(6) Before equation 28, the authors write D=-2. Do they mean D=-2U? Furthermore, it would be better to explicitly name the long-range order as m3 and m4.

(7) (Minor) This might be only my feeling, but my first thought when reading equation 18 in a section called DYNAMICAL Hartree approximation was that t is the time. However, t seems to be the hopping. It would be good to clarify this.

---

## Round 1 · Referee Report · Anonymous (Referee 2) · 2022-1-23

Strengths

1) Application of a well-suited technique to study symmetry breaking orders in a model with non-local interactions

Weaknesses

1) The cited literature is incomplete 2) The study does not include the simultaneous breaking of spin/charge symmetry via appropriate Weiss fields and limits itself to one type of magnetic and charge order respectively

Report

In this paper, the authors study the extended Hubbard model on a two-layer honeycomb lattice as introduced to describe twisted bilayer graphene (TBG) by means of the variational cluster approximation (VCA). The employed cluster technique is adequate to address the question of symmetry breaking in correlated electron systems and in combination with the dynamical Hartree approximation even non-local interactions can be -to some extend- included.
The VCA is applied to a 12-site cluster on which local and inter-site correlations are taken into account exactly. Extended inter-cluster interactions are decoupled within dynamical Hartree approximation and long-range order is allowed for by cluster Weiss fields whose strength is determined via a variational principle.
The focus is set on the Mott transition at half- and quarter filling as well as on antiferromagnetic and charge density instabilities for the respective fillings. In both cases, the authors find absence of the symmetry-breaking solutions if the extended interactions of the model obey a given set of relations (eq. (5) of the paper).
It is then concluded that for these filling ratios the ground state of the model is of Mott insulating type and not gapped due to symmetry breaking in the spin or charge sector.

The paper is interesting and can constitute an important contribution to help to pinpoint the importance of non-local interactions and correlation effects on the insulating nature of TBG at half- and quarter filling.
However, the current version of the manuscript includes a few weaknesses which need to be corrected before I can recommend it for publication in SciPost Physics.
In particular, the citations and discussion of the relevant literature need to be updated to meet the journal's acceptance criteria.

Requested changes

The following points should be addressed:

1) A proper comparison to the existing literature on TBG both from theory and experiment is missing. In particular, the results need to be compared to other theory papers that investigate the importance of non-local interactions for the ground state properties at half- and quarter filling. Even more studies exist for the Hubbard model with on-site interactions applied to TBG. By mean-field-decoupling the non-local interaction terms, the authors could introduce an effective U to compare to those papers, too.

2) One of the issues of cluster techniques is always the analysis of finize-size effects. Since the 12-site cluster is close to the limit of numerically exploitable cluster sizes, a full finite-size scaling is not feasible. Still, the comparison to at least one additional cluster size/geometry would be helpful. One such candidate could be a supercluster of 8-site clusters (4 sites x 2 layers), which was already employed within VCA in similar contexts.

3) Another point concerns one of the strenghts of VCA, which is not used to full capacity here, namely the possibility to check for the competition of different symmetry-breaking fields on equal footing. For instance, at quarter filling it would be important to check for breaking of spin- and charge-order, as it has been discussed in literature for TBG at different magic angles.

4) When explaining the Hartree decoupling of the inter-cluster interactions, the authors cite Refs. 16 & 18. However, a reference to the first paper that introduced this type of mean-field decoupling in context of VCA is missing and needs to be added: PRB 70, 235107 (2004).

5) The authors explain how to determine the mean-fields in the dynamical Hartree approximation. In the present case, is there a specific reason why the authors decided to choose the variational determination of these fields over a self-consistent determination?

6) A central question for the applicability of the studied model to TBG concerns the values and structure of the interaction terms. The choice of the interactions (e.g. the relations (5) and the considered values) needs to be discussed properly. In their previous study, Ref. 7, the authors devoted a small paragraph to explaining their choice of values of U. Here, such an analysis is even more important. In particular, it should be discussed in how far the considered interactions V0-V3 agree with ab initio calculations of the screened interactions (cRPA, e.g. Refs. cited in Ref.7, or self-consistent atomistic Hartree theory, PRB 103, 195127 (2021)).

7) How robust is the absence of AF order at half-filling with respect to the choice of U when deviating from the case a=1? Is the 'critical' value of a changing with U?

8) The authors studied AF order at half-filling, but other magnetic orders are discussed in context of TBG for different fillings. Can the authors exclude other magnetic orders (e.g. FM order or on-site AF ordering between the two layers) at half- or quarter filling within their VCA setup?

Finally, some minor points:

9) Throughout the manuscript it should be specified in which units U is measured (in meV and not in units of the largest hopping?).

10) The matrix of differences between the one-body terms of the lattice and the reference system is called V, see e.g. eq(17). Although being standard nomenclature in context of VCA, this naming is slightly inept here since it can be easily confused with the interaction terms V, see e.g. eq(18).

---

## Round 1 · Referee Report · Anonymous (Referee 3) · 2022-1-24

Strengths

1- Interesting subject and which is a hot topic of the condensed matter of 2D materials.

2- Advanced numerical method adapted to complexity of the studied system

Weaknesses

1- This topic has been the subject of many publications since its appearance in 2018. I understand that the authors can not quote all the articles that have been published on it. However, a state of the art is missing and the comparison of the results with the recent articles dealing with the subject seems necessary.

2- The results need to be a little more detailed.

3- These results are obtained with a particular tight binding model developed for systems without interaction. It is not at all obvious that this model is valid with interactions. Of course this is often the case, which is why it is important to discuss the validity of the model used.

Report

In this manuscript, B. Pahlevanzadeh, P. Sahebsara, D. Sénéchal study the effect of electronic interactions in magic-angle twisted bilayer graphene. They focus on quarter- and half-filling using a tight binding model proposed by Kang and Vafek (Ref. [1]) to simulate the 4 low-energy bands. For these numerical calculations they use the variational cluster approximation (VCA), which allows to include extended interactions.
The quality of the work seems to me good and the method well adapted for this study. The methods are well presented in the manuscript. The main conclusions too, however I find that the results could be a bit more detailed.
This work brings undoubtedly new results in a very active field and that is why I think it could be published in SciPost Physics, however it is to me necessary that the authors answer the different remarks and questions to be able to give a definitive opinion.

Requested changes

1- Update the manuscript by taking into account remarks of section “Weaknesses”.

2- The authors have recently published an article in SciPost Physics with the same model (Ref. [7] in the present manuscript). The 2 articles are not redundant. However, the authors should justify explicitly the need for a new article and explain the overall coherence of their work based on the model proposed by Kang and Vafek (Ref. [1]).

3è The figures 1 and table 1 seem to be exactly the same as the figure 1 and table 1 of the Ref. [7], is it necessary to show them again ?

4- In this work, the authors use a 12-site cluster containing 3 unit cells. Is it possible to justify this choice?

5- The strong-coupling limit is presented in section 2.1. Although this limit is interesting for itself, I do not think it is applicable to the case of magic-angle twisted bilayer graphene. Indeed in the strong coupling limit, a 4 bands model is not sufficient because the interactions will have also a strong effect on many more bands.

6- Page 10, it is written: “This Mott transition is essentially caused by extended interactions”. The authors should elaborate a bit more on this point and complete it.

7- Section 5, antiferromagnetism is shown at half filling for not to strong interaction. Can the authors specify the spatial magnetization state? Is the antiferromagnetism found for the interlayer order, the interlayer order or both?

---

## Round 2 · Referee Report · Anonymous (Referee 3) · 2022-5-17

Report

The authors have replied to all questions of the referees.
Although I think that the manuscript would have benefitted from a finite-size scaling, as one of the referees proposed, I accept the author's answer.
I recommend this manuscript for publication in SciPost Physics.

---

## Round 2 · Referee Report · Anonymous (Referee 2) · 2022-5-21

Report

The authors replied thoroughly to all remarks and answered all pertinent questions of the referees. The current version of the manuscript is much improved, in particular since it now includes a discussion of possible FM order and the problems related to the discontinuous behavior of the self-energy functional.
I therefore recommend the publication of this article in SciPost Physics.

Requested changes

Below, I list a few typos that the authors might want to correct for the final version.
- p.2, 1st line of section 2: "[...] tight-binding Hamiltonians proposed [...]"
- p.13, caption of Fig.7 : "Right panel: same as left panel, but [...]"
- p.14 : "[...] absence of antiferromagnetism extends to [...]"
- References: Please check for upper-case letters in the titles ("Mott", "Coulomb" etc.) and use a consistent nomenclature for the journals (e.g. either "Phys. Rev. Lett." or "Physical Review Letters"); typo in Ref.24 ("[...] mean-field theory").

---

## Round 2 · Referee Report · Anonymous (Referee 1) · 2022-5-23

Report

The authors have submitted a new version with a significant number of additions and updates. They have satisfactorily answered my questions and, I believe, those of the other referees. Following my first report, I therefore consider that this manuscript deserves to be published in SciPost Physics.

---

## Round 2 · Author Response

RESPONSE TO REFEREE REPORTS

The referees all bring excellent points and suggestions, some of them overlapping. In the space below we quote the various change requests made by the referees, with our response immediately below (following the RESPONSE keyword). Reference numbers between brackets [...] correspond to references at the end of the new version of the paper (references cited in the referee reports have been renumbered to take this into account).

Changes demanded by Report # 3

1) Update the manuscript by taking into account remarks of section “Weaknesses”(detailed below):

A) This topic has been the subject of many publications since its appearance in 2018. I understand that the authors can not quote all the articles that have been published on it. However, a state of the art ismissing and the comparison of the results with the recent articles dealing with the subject seems necessary. B) The results need to be a little more detailed. C) These results are obtained with a particular tight binding model developed for systems without interaction. It is not at all obvious that this model is valid with interactions. Of course this is often the case,which is why it is important to discuss the validity of the model used.

RESPONSE Various changes to the manuscript and many responses below address this remark. The literature on the treatment of interactions in TBG has been better cited.

2) The authors have recently published an article in SciPost Physics with the same model (Ref. [24] in the present manuscript). The 2 articles are not redundant. However, the authors should justify explicitly the need for a new article and explain the overall coherence of their work based on the model proposed by Kang and Vafek (Ref. [1]).

RESPONSE In the introduction, We augmented the precise justification for this paper, compared to Ref.[24] : I boils down to the need for a larger cluster, in order to have a more dynamical (less mean-field) contributions from the extended interactions within the cluster in order to study the normal phase. This in turns prevents us from using the method used in Ref.[24] (CDMFT) and to use instead the variational cluster approximation.

3) The figures 1 and table 1 seem to be exactly the same as the figure 1 and table 1 of the Ref. [24], is it necessary to show them again ?

RESPONSE Fig. 1 and Table 1 are indeed borrowed from our previous work on SciPost and are reproduced here to facilitate reading. This is stated in the caption. We felt that an in electronic medium such as SciPost this would not incurr extra cost but would benefit the reader. We can replace these figures by mere references is the Editors prefer it that way.

4) In this work, the authors use a 12-site cluster containing 3 unit cells. Is it possible to justify this choice?

RESPONSE The 12 site cluster is the largest one we can treat that has the symmetry of the model. It allows us to treat a fair fraction of the extended interactions within the cluster (and thus capture the dynamical correlations) and at the same time all sites on it are equivalent by symmetry, which simplifies the Hartree approximation for the inter-cluster interactions. This comment has been added to the introduction.

5) The strong-coupling limit is presented in section 2.1. Although this limit is interesting for itself, I do not think it is applicable to the case of magic-angle twisted bilayer graphene. Indeed in the strong coupling limit, a 4 bands model is not sufficient because the interactions will have also a strong effect on many more bands.

RESPONSE The referee is correct. We use this limit not as applying to TBG itself, but as a useful prelude to the interacting model we use, as a guide for its solution. We added a comment to this effect in the manuscript. The strong-coupling limit has been covered in a few references [23, 13, 10], albeit not in the way we do, but I believe in the same spirit.

6) Page 10, it is written: “This Mott transition is essentially caused by extended interactions”. The authors should elaborate a bit more on this point and complete it.

RESPONSE We have added a few sentences to elaborate on this point.

7) Section 5, antiferromagnetism is shown at half filling for not too strong interaction. Can the authors specify the spatial magnetization state? Is the antiferromagnetism found for the interlayer order, the interlayer order or both?

RESPONSE It is Néel antiferromagnetism, sublattice based and intra-layer only. Eq. (28) states it thus, but we agree this is not clear enough and we have added sentences to clarify. We have also added the possibility of both inter-layer and intra-layer antiferromagnetism. It turns out that this makes no difference, owing to the small value of the inter-layer hopping.

Changes demanded by Report # 2

1) A proper comparison to the existing literature on TBG both from theory and experiment is missing. In particular, the results need to be compared to other theory papers that investigate the importance of non-local interactions for the ground state properties at half- and quarter filling. Even more studies exist for the Hubbard model with on-site interactions applied to TBG. By mean-field-decoupling the non-local interaction terms, the authors could introduce an effective U to compare to those papers, too.

RESPONSE The literature on the treatment of interactions in TBG has been better cited (this is also in the introduction). Also, comparing critical interaction values with other works is a challenge, because of the differences in the models used and/or twist angle. In Ref. [12], the critical U for the Mott insulator at quarter filling is 14.7t. But their non-interacting model is quite different (2 bands, with nearest-neighbor hopping t = 2meV only), as well as their method of solution (slave bosons). This puts their critical U at about 29 meV, much larger than our critical U of 1.5-2 meV! They do state, however, that extended interactions, which they do not take into account, would lower that value. Critical interaction strengths are also discussed in [14], but in the context of a microscopic model, not an effective model, making any comparison near impossible (the energy scales are in eV, not meV !)

2) One of the issues of cluster techniques is always the analysis of finize-size effects. Since the 12-site cluster is close to the limit of numerically exploitable cluster sizes, a full finite-size scaling is not feasible. Still, the comparison to at least one additional cluster size/geometry would be helpful. One such candidate could be a supercluster of 8-site clusters (4 sites x 2 layers), which was already employed within VCA in similar contexts.

RESPONSE Performing the same computation on a smaller cluster such as the 8-site cluster suggested by the referee is in fact more difficult than on the 12-site cluster, because the 8-site cluster is based on the 4-site star-shape cluster that contains a center site that is not equivalent to the edge sites. Thus additional care must be taken in the Hartree approximation: More mean-field terms must be added and the simple charge-density wave patterns studied on the 12-site cluster correpond to complicated mixtures of inter-cluster Hartree fields and intra-cluster Weiss fields. This is why we do not carry out these computations.

3) Another point concerns one of the strengths of VCA, which is not used to full capacity here, namely the possibility to check for the competition of different symmetry-breaking fields on equal footing. For instance, at quarter filling it would be important to check for breaking of spin- and charge-order, as it has been discussed in literature for TBG at different magic angles.

RESPONSE Some authors expect a ferromagnetic state at quarter or three-quarter filling (e.g. Ref [15]). Ferromagnetism has been likely detected in TBG with a twist angle of about 1.20 degree at 3/4 filling [38]. It is difficult to use VCA to look for ferromagnetism at 1/4 filling, on top of the insulating state found. We explain why in the revised manuscript. However, an analysis of the low-lying states of the cluster shows that it is nearly ferromagnetic. In addition, a state that is ferromagnetic within layers and antiferromagnetic between layers was not found with VCA (the minimum of the Potthoff functional is at zero). A paragraph and a figure to that effect were added to the manuscript.

4) When explaining the Hartree decoupling of the inter-cluster interactions, the authors cite Refs. 16 & 18. However, a reference to the first paper that introduced this type of mean-field decoupling in context of VCA is missing and needs to be added: PRB 70, 235107 (2004).

RESPONSE An obvious omission, for which we apologize. We added this reference [36].

5) The authors explain how to determine the mean-fields in the dynamical Hartree approximation. In the present case, is there a specific reason why the authors decided to choose the variational determination of these fields over a self-consistent determination?

RESPONSE In ordinary mean-field theory, the mean-field can be determined either by minimizing the free energy or by applying self-consistency; the result is the same. In the dynamical Hartree approximation, this is not obviously the case. We feel the variational approach to be superior because of its presumed stability. Self-consistent procedure may sometimes diverge even if a solution exists. In the case of the charge order at quarter filling, we checked that the self-consistent approach also converged to zero (no charge order) and added a remark to that effect in the manuscript.

6) A central question for the applicability of the studied model to TBG concerns the values and structure of the interaction terms. The choice of the interactions (e.g. the relations (5) and the considered values) needs to be discussed properly. In their previous study, Ref. 7, the authors devoted a small paragraph to explaining their choice of values of U. Here, such an analysis is even more important. In particular, it should be discussed in how far the considered interactions V0-V3 agree with ab initio calculations of the screened interactions (cRPA, e.g. Refs. cited in Ref.7, or self-consistent atomistic Hartree theory, PRB 103, 195127 (2021)).

RESPONSE In Fig. 10 of Ref. [29], the ratios of 1st and 2nd neighbor interactions to the local interaction are 12/28 and 9/28, instead of 2/3 and 1/3. This is in close agreement for the 2nd neighbor interactions, but 33% off for the first-neighbor interactions. In principle this depends on the twist angle (see also Table II of [29]). We commented on this in the revised version of the paper. Klebl et al, [10.1103/PhysRevB.103.195127] deals with a microscopic model, not a moiré tight-binding effective model, so the comparison is difficult.

7) How robust is the absence of AF order at half-filling with respect to the choice of U when deviating from the case a=1? Is the 'critical' value of a changing with U?

RESPONSE A plot for a=1 and various values of U has been added. The critical value of 'a' may depend on U, but antiferromagnetism at half filling is absent for the range of U studied in that plot.

8) The authors studied AF order at half-filling, but other magnetic orders are discussed in context of TBG for different fillings. Can the authors exclude other magnetic orders (e.g. FM order or on-site AF ordering between the two layers) at half- or quarter filling within their VCA setup?

RESPONSE In VCA, like in mean-field theory, one cannot exclude orders that are not explicity probed. At half-filling, there is no significant difference between antiferromagnetism and ferromagnetism accross computational layers, owing to the small value of the interlayer hopping term (in both case we probe Néel antiferromagnetism within each layer). See response to comment 3.7 above. We have also made remarks about ferromagnetism at quarter filling. See response to comment 2.3 above.

9) Throughout the manuscript it should be specified in which units U is measured (in meV and not in units of the largest hopping?).

RESPONSE All parameters are in meV. This is not clear enough and the manuscript has been modified accordingly, especially in the figures.

10) The matrix of differences between the one-body terms of the lattice and the reference system is called V, see e.g. eq(17). Although being standard nomenclature in context of VCA, this naming is slightly inept here since it can be easily confused with the interaction terms V, see e.g. eq(18).

RESPONSE We agree. We have changed the notation for the inter-cluster one-body matrix.

Changes demanded by Report # 1

1) Although there are many publications about correlation effects in twisted bilayer Graphene in experiment and theory, the authors cite only 18 publications. This must be expanded. A short and not extensive list would be PhysRevX.8.031089 PhysRevB.98.081102 J. Phys. Commun. 3 035024 PhysRevLett.124.097601 PhysRevB.102.035136 PhysRevB.102.045107 PhysRevB.102.085109 SciPostPhys.11.4.083 PhysRevB.100.155145

As several of these papers also discuss Mott states and antiferromagnetic states, the current results should be compared to these previous results.

RESPONSE The literature on the treatment of interactions in TBG has been better cited (this is also in the introduction).

2) During the explanation of the model, I am missing the spin degrees of freedom, which suddenly appear in equation 4. This should be expanded. Furthermore, the condition on the spin degrees should be stated in the strong-coupling section and the calculations for quarter-filling.

RESPONSE The presence of spin is insisted upon from the beginning of Sect. 2 in the new version of the manuscript.

3) Maybe I misunderstood something in the calculation of the strong-coupling limit. When the largest eigenvalue corresponds to the uniform state, then the lowest eigenvalues correspond to nonuniform (charge-ordered) states. Why is the ground state in the strong-coupling limit (when neglecting the kinetic energy) not long-range ordered?

RESPONSE Good question! In fact the largest eigenvalue is related to the total charge, which is conserved and cannot be changed unless the chemical potential is changed. This uniform mode being constrained by particle number conservation, it can only serve as a background with respect to which the other (non uniform) charge modes are playing out. The ground state would be a charge density wave if one of the modes had a negative eigenvalue. For instance, refereing to Eqs (14-16), in the eventuality where $V_0=V_2=V_3=0$ and $V_1$ is nonzero, there would be an inter-orbital charge density wave at $q=0$ if $V1 > U/2$, as the eigenvalue $\lambda^{(2)}$ would then turn negative. But, as stated in the paper, under the conditions (5), this is not possible. We have added additional explanations in the last paragraph of Sect. 2 to clarity this.

4) In the section about VCA, the details about how the finite cluster has been solved are missing. This should be expanded.

RESPONSE A few sentences have been added about the impurity solver used.

5) The authors analyze at quarter-filling only charge order and at half-filling only antiferromagnetism. Why is there no analysis of combinations of charge and spin order? Furthermore, other long-range spin orders other than antiferromagnetism should be discussed. If possible, the authors should include combined spin and charge order calculations in this manuscript.

RESPONSE See our response to comments 2.3 and 2.8 above. Regarding orders mixing spin and charge, a complete analysis of this question would go beyond the scope of this paper. The possibility of stripe magnetic order is interesting, but is naturally expected outside of the special fillings studied here.

6) Before equation 28, the authors write D=-2. Do they mean D=-2U? Furthermore, it would be better to explicitly name the long-range order as m3 and m4.

RESPONSE Indeed, $D=-2U$. Thank you for picking this up.

7) (Minor) This might be only my feeling, but my first thought when reading equation 18 in a section called DYNAMICAL Hartree approximation was that t is the time. However, t seems to be the hopping. It would be good to clarify this.

RESPONSE Indeed. It is called dynamical because it is coupled to a variational principle involving the Green function. This has been clarified in the revised version.

---

## Round 2 · List of Changes

The list of changes appears together with the response to the referee reports. A latexdiff file is also provided immediately after the new version of the manuscript (blue passages are additions, red passages were removed).

---

## Editorial Decision

published